# The Early Dynamic Change in Cardiac Enzymes and Renal Function Is Associated with Mortality in Patients with Fulminant Myocarditis on Extracorporeal Membrane Oxygenation: Analysis of a Single Center’s Experience

**DOI:** 10.3390/healthcare10061063

**Published:** 2022-06-08

**Authors:** Ching-Lin Ho, Teressa Reanne Ju, Chi Chan Lee, Hsin-Ti Lin, Alexander-Lee Wang, Robert Jeenchen Chen, You-Cian Lin

**Affiliations:** 1Department of Surgery, Division of Cardiovascular Surgery, China Medical University Hospital, Taichung City 406040, Taiwan; danielho196@gmail.com; 2Department of Internal Medicine, New York Presbyterian Queens, New York, NY 11355, USA; tej9016@nyp.org; 3Department of Critical Care, Guam Regional Medical City, Tamuning, GU 96929, USA; ccl1985@gwu.edu; 4College of Medicine, China Medical University, Taichung City 406040, Taiwan; ccindylin@gmail.com (H.-T.L.); a.w.rhhs@gmail.com (A.-L.W.); 5Department of Cardiothoracic Surgery, Stanford University School of Medicine, Palo Alto, CA 94305, USA; dr.rjcc@gmail.com

**Keywords:** fulminant myocarditis, extracorporeal membrane oxygenation (ECMO), cardiac enzymes, renal function

## Abstract

(1) Background: Fulminant myocarditis (FM) could result in hemodynamic derangement and fatal arrhythmia. Veno-arterial extracorporeal membrane oxygenation (V-A ECMO) is used to maintain organ perfusion in FM patients complicating cardiogenic shock. The present study aims to assess the static and dynamic factors in association with mortality in FM patients on V-A ECMO (2) Methods: Twenty-eight patients were enrolled between 2013 to 2019 for analysis (3) Results: In-hospital survival rate was 78.5%. There was no statistical difference in demographics and baseline laboratory data between survivors and non-survivors. However, within 24 h after ECMO support, CK-MB increased by 96.8% among non-survivors, but decreased by 23.7% among survivors (*p* = 0.022). Troponin I increased by 378% among non-survivors and 1.7% among survivors (*p* = 0.032). Serum creatinine increased by 108% among non-survivors, but decreased by 8.5% among survivors (*p* = 0.005). The receiver operating characteristic curve suggested an increase in serum creatinine by 68% within 24 h after ECMO support was associated with increased mortality with an area under the curve of 0.91. (4) Conclusions: V-A ECMO is an excellent tool to support FM patients with cardiogenic shock. The early dynamic change of renal function and cardiac enzymes may be useful for outcome assessment.

## 1. Introduction

Fulminant myocarditis (FM) is a rare and distinct form of myocarditis complicating hemodynamic derangement and fatal arrhythmia [1]. In patients who develop cardiogenic shock secondary to fulminant myocarditis, despite the use of inotropic agents, mechanical circulatory support may be needed to maintain end-organ perfusion [2]. In contrast to ventricular assist device (VAD), which requires open-heart surgery, veno-arterial extracorporeal membrane oxygenation (V-A ECMO) has emerged as a salvage therapy readily available to support cardiopulmonary function during the acute phase of illness [3]. A number of studies [4,5,6,7,8,9,10,11,12] have reported their experiences of using V-A ECMO in patients with FM. Two studies [7,8] investigated the prognostic factors for myocardial recovery, and two studies [5,6] investigated the prognostic factors for survival. While previous studies assessed prognostic factors using variables at specific time points (e.g., peak troponin-I level 24 h before ECMO), little is known about the implication of the dynamic change of organ functions before and after ECMO support in outcome prognostication. The aim of our study is to examine the static as well as dynamic factors in association with mortality in patients with FM on V-A ECMO.

## 2. Materials and Methods

A retrospective chart review was conducted to include patients who were diagnosed with acute FM at the China Medical University Hospital (Taichung, Taiwan) from January 2013 to December 2019. There was no reported COVID-19 case during this period in Taiwan. The study received an exemption determination from the Institutional Review Board of China Medical University Hospital (CMUH109-REC1-017) since the study was in the category of non-interventional clinical research, which was designed to review clinical data retrospectively.

### 2.1. Population

The included patients met all the criteria: (1) age ≥ 18 years old, (2) onset of disease less than two weeks, (3) documentation of recent symptoms of upper respiratory tract infection or gastrointestinal infection, (4) severe left ventricular dysfunction on echocardiography, (5) elevated cardiac enzymes, (6) no acute coronary artery occlusion on cardiac catheterization, and (7) hemodynamic instability needing ECMO support. Evidence from a myocardial biopsy was not required in the present study. The exclusion criteria were: (1) history of transplantation or left VAD, and (2) history of heart failure or myocardial infarction.

### 2.2. Device and Management

The peripheral V-A ECMO system consisted of a centrifugal pump, a microporous polypropylene hollow fiber membrane oxygenator, and a heparin-coated circuit (Medtronic Inc., Minneapolis, MN, USA). A Rota Flow pump with a polycarbonate membrane oxygenator and bioline-coated circuit (Maquet Inc., Rastatt, Germany) would be used as second-line therapy if the oxygenator failed. We used a 15–19 French (Fr) arterial cannula with a length of 15–18 cm and a 17–21 Fr venous cannula with a length of 50–60 cm, both of which were inserted via the femoral vessels using the puncture approach. Mechanical circulation was established with venous blood drainage from the right atrium and arterial blood return to the iliac artery.

The management and weaning of ECMO were performed according to the published guideline [13]. Low-dose heparin was continuously administered to maintain an activated clotting time of 160–200 s. To prevent poor venous drainage, right atrium pressure was maintained equal to or above 12 mmHg with adequate fluid administration, and blood transfusion was performed to maintain a hemoglobin level equal to or above 10 g/dL and a platelet level equal to or above 80,000/μL. A distal perfusion catheter into the superficial femoral artery was routinely used to prevent distal leg ischemia. The initial flow rate was 3.0–4.0 L/min to assist recovery from peripheral circulatory failure. In all patients, ECMO was introduced in intensive care units, emergency departments, or catheter laboratories. The flow rate of ECMO was weaned based on the indicators of organ perfusion (e.g., arterial blood gas analysis, mixed venous oxygen saturation, lactic acid, and urinary output). When the flow rate reached 0.8 L/min, echocardiography showed evidence of heart recovery, and if the indicators mentioned above were acceptable, ECMO could be weaned off.

### 2.3. Data Collection and Statistical Analysis

In-depth chart reviews were conducted independently by two medical interns and one fellow physician from the cardiovascular surgery department. Demographic data, such as age, gender, body mass index, and comorbidities were collected at the time of admission. Survival after the veno-arterial ECMO (SAVE) score [14], a score developed by the Extracorporeal Life Support Organization (ELSO) to assist the prediction of survival for adult patients receiving ECMO for refractory cardiogenic shock, was calculated. Laboratory data at 24 h before, 24 h after, and 24 to 48 h after ECMO cannulation were recorded. If multiple data existed during the same period, the data point with the highest value was used for analysis. The use of steroid or intravenous immunoglobulin (IVIG) was recorded based on prescription orders of electronic medical records. Time to ECMO cannulation was defined as the duration from emergency department (ED) triage to ECMO cannulation according to provider notes and procedure notes. An echocardiogram was performed within 24 h of ECMO support. Regarding the outcome variables, in-hospital mortality rate, duration of ECMO support, ICU length of stay, and need for renal replacement therapy were reported. The definition of major bleeding referred to the bleeding that required equal or more than four units of packed red blood cell transfusions or procedural interventions. Missing data were documented and reported in the tables. Variables that had more than five missing data at any time point were reported and excluded for data analysis.

Categorical variables were mostly analyzed with Fisher’s exact test since there were only six non-survivors in the cohort and expected frequencies were mostly less than or equal to five. For continuous variables, histogram visualization and the Shapiro–Wilk test were utilized to determine the patterns of variable distributions. Continuous variables with normal distributions were expressed as the mean ± standard deviation and analyzed with independent Student’s *t*-test, while those without normal distributions were expressed as the median (interquartile range (IQR), 25–75%) and analyzed with the Mann–Whitney *U* test. A two-sided *p*-value of 0.05 or lower was considered statistically significant. A receiver operating characteristic (ROC) curve was conducted to analyze the cutoff value of difference of variables 24 h before and after ECMO support. All statistical analysis was conducted using Stata software, version 17 (StataCorp, College Station, TX, USA).

## 3. Results

Between January 2013 and December 2019, a total of 28 patients were diagnosed with fulminant myocarditis and received ECMO support. All patients received cardiac catheterization during hospitalization to rule out acute coronary syndrome (ACS). Six patients died while receiving ECMO support. Among the 22 survivors, 15 patients had a recovery of heart function and underwent successful ECMO decannulation. Two patients received VAD placement and five received cardiac transplantation. The in-hospital survival rate of the cohort was 78.5% (Figure 1).

Table 1 shows the patient characteristics, SAVE score, use of IVIG or steroid, left ventricular ejection fraction (LVEF), and laboratory data. There was no statistical difference in the demographics and comorbidities between survivors and non-survivors. SAVE scores, while lower in non-survivors than survivors, did not reach statistical difference (−4.3 ± 6.5 versus −1.4 ± 5.8, *p* = 0.29). There was no significant difference in the percentage of patients who received steroid or IVIG, LVEF (19.9% versus 23.1%), and laboratory data at baseline between non-survivors and survivors. The median peak troponin-I level during hospitalization was 66.6 ng/mL (16.6–82) in the non-survivor group and 31.9 ng/mL (10.9–47.6) in the survivor group (*p* = 0.15). The median time of ECMO cannulation from ED triage was 681 (140–1260) minutes in the non-survivor group and 315 (140–766) minutes in the survivor group (*p* = 0.73).

Table 2 shows the clinical outcomes and incidences of major complications. The median duration of ECMO support was 4.5 (IQR: 4–9) days in the non-survivor group and 8 (IQR: 5–14) days in the survivor group (*p* = 0.1). Renal replacement therapy was applied to 83.3% of non-survivors and 45.5% of survivors (*p* = 0.23). ICU length of stay was significantly longer in survivors than non-survivors (20 versus 5 days, *p* = 0.002). In terms of major complications, one patient developed limb ischemia and one patient developed bowel ischemia in the non-survivor group. One patient had major bleeding from the ECMO cannula insertion site, which required procedural interventions in the survivor group. The incidence of major GI bleeding was similar between the two groups (16.7% versus 18.2%, *p* = 1). No ECMO cannula malposition or stroke was observed in the cohort.

Figure 2 shows the laboratory data at different time points before and after ECMO support. While there was no difference in the laboratory data at baseline, compared to the survivors, non-survivors had significantly higher creatinine [3.2 (IQR: 2.3–3.6) mg/dL versus 1.5 (IQR: 0.7–2.5) mg/dL, *p* = 0.026] at 24 to 48 h after ECMO support and CPK [4762 (IQR: 1492–19,422) IU/L versus 855 (IQR: 468–2332) IU/L, *p* = 0.04] at 24 h after ECMO support.

Table 3 shows the dynamic change in laboratory data within 24 h before and after ECMO support between non-survivors and survivors. Within 24 h after ECMO support, the CK-MB increased by 96.8% (IQR: 23.3–144.1) among non-survivors but decreased by 23.7% (IQR: −41.5 to −10.4) among survivors, and the change in CK-MB levels was significantly different between the two groups (*p* = 0.022). Troponin I increased by 378% (IQR: 144 to 1359.9) among non-survivors and only increased by 1.7% (IQR: −18 to 147.5) among survivors. The change in troponin-I level was also statistically different between the two groups (*p* = 0.032). In terms of kidney function, there were statistical differences in the change of BUN [non-survivors: 46.4% (IQR: 33.3 to 61.9); survivors: 3.9% (IQR: −23.6 to 18.5), *p* = 0.006] and creatinine [non-survivors: 108% (IQR: 52.3 to 156.6); survivors: −8.5% (IQR: −25.6 to 17.5), *p* = 0.005] within 24 h after ECMO support between the two groups.

ROC curves were created and fitted using non-linear binomial regression. Differences of CK-MB, troponin-I, and creatinine levels 24 h before and after ECMO support had an area under the curve (AUC) of 0.74, 0.8, and 0.89 to predict in-hospital mortality, respectively (Figure A1). Based on the non-parametric estimation ROC curve using Delong’s method (Figure 3), serum creatinine increased by 68% after ECMO support was associated with higher mortality with an AUC of 0.91 (sensitivity: 83.3%, sensitivity: 92.3%, positive likelihood ratio (LR): 10.8, negative LR: 0.1806; Table A1).

## 4. Discussion

The present study assessed factors in association with in-hospital survival in patients with FM on V-A ECMO. At baseline, there was no significant difference between non-survivors and survivors in demographics, SAVE score, LVEF, and laboratory data. However, changes in cardiac enzymes and renal function before and after ECMO support were significantly different between non-survivors and survivors, indicating the potential advantage of assessing dynamic change in organ functions as early as 24 h after ECMO support for outcome prognostication. 

The survival rates of patients who had FM on V-A ECMO reportedly ranged from 59% to 83% [4,5,6]. The differences in mortality across studies could be explained by the differences in demographics and percentage of patients who received CPR. For example, Nakamura et al. [5] reported the survival rate for patients who had FM on V-A ECMO was 59%. The mean age of non-survivors was 60.2 years, which was higher than most studies. Hsu et al. [4] reported the survival rate of patients to be 60.8%. Interestingly, 47% of the included patients received CPR during hospitalization. In our cohort, the in-hospital survival rate was 78.5%. The included population in our study was younger with fewer comorbidities. Evidently, the factors associated with poor survival included older age [5], complications following ECMO placement [5], and high lactate level prior to ECMO support [6]. In our study, we did not observe differences in the demographics, complication rates, SAVE score, LVEF, and laboratory data at baseline between non-survivors and survivors. This phenomenon may be due to the small sample size of our study, in particular, only six patients were identified in the non-survivor group. It is worth mentioning that the SAVE score does incorporate myocarditis as one of its protective factors in the score system. Since no previous study specifically assessed the utility of the SAVE score in patients with FM, further studies are needed to determine the validation of using the SAVE score for this population.

In our cohort, the duration of ECMO support among survivors was 8 days, which was similar to the 6.6 days in the study of Hsu et al. [4] and 7.5 days in the study of Nakamura et al. [5]. There were no patients who required emergent VAD surgery or restarting ECMO support following the explantation of ECMO. This phenomenon may be explained by the small sample size or our team being overly cautious in ECMO explantation. The optimal timing of bridging therapy was rarely studied and could be a factor associated with the clinical outcomes. In our practice, we generally waited for 10 to 14 days before initiating the arrangements for bridging therapy. Further studies are needed to delineate the time course of disease development so that physicians can arrange bridging therapy in a timely manner and reduce the duration of ECMO support and hospital stay.

In our study, most patients did not receive IVIG or steroids. The roles of immunosuppressive therapies in treating patients with FM remain controversial and vary depending on the etiology of the disease. Previous studies [15,16] showed IVIG had potential benefits in reducing viral load and improving LVEF. For giant cell and eosinophilic myocarditis, a high-dose steroid has shown to improve mortality by attenuating inflammation of the myocardium [1]. On the other hand, large randomized controlled trials (RCTs) failed to show survival benefit or improvement of left ventricular recovery in patients with acute myocarditis and an LVEF less than 45% [17] or new-onset idiopathic dilated cardiomyopathy/myocarditis [18]. A recent nationwide retrospective cohort study in Japan revealed that IVIG use in adult patients with FM was not associated with a decrease in in-hospital mortality [19]. It may be reasonable to empirically apply immunosuppressive therapies to critically ill patients secondary to FM because some of them could have diseases, such as giant cell or eosinophilic myocarditis, which may respond to treatment. Future RCTs are needed to identify the populations that will benefit from IVIG and steroids, preferably with criteria or laboratory evidence that are less invasive than tissue biopsy.

Cardiac enzymes are biomarkers reflecting the extent of myocardial injury and may be associated with clinical outcomes in patients with acute myocarditis [2]. Chou et al. [20] revealed that peak CK-MB level was one of the key prognostic factors of cardiac function recovery. Chong et al. [11] observed higher peak troponin-I levels in the non-survivor group. In our study, there was no difference in the peak troponin-I (*p* = 0.15) and peak CK-MB (*p* = 0.16) levels between non-survivors and survivors. However, within 24 h of ECMO support, non-survivors had troponin-I levels increased by 96.8% and CK-MB levels increased by 378%, while survivors had troponin-I levels increased by 1.7% and CK-MB levels decreased by 23.7%. V-A ECMO could facilitate ventricular recovery by decreasing wall tension, improving coronary perfusion, and reducing the need of inotropes, which are arrhythmogenic [21]. Therefore, a significant worsening of cardiac enzymes shortly after ECMO support may imply profound myocardial injury, despite restoration of cardiac output. Compared to the level of cardiac enzymes presented at specific time points, changes in cardiac enzymes after ECMO support may be more sensitive and indicative of poorer clinical outcomes.

Severity of end-organ damage was previously studied in patients with FM on V-A ECMO. Diddle et al. [12] reviewed the ELSO registry database and found that, among 147 patients who had acute myocarditis on V-A ECMO, central nervous system injury, acute renal failure, and hyperbilirubinemia were individually associated with increased odds of in-hospital mortality. Nakamura et al. [5] reported that the serum bilirubin level on day 7 was significantly lower in the survivor group. In our study, we noticed that, following ECMO support, creatinine at 24–48 h and CPK at 0–24 h were significantly higher in non-survivors. Unfortunately, we did not have enough data to analyze the differences of bilirubin levels and liver function enzymes between the two groups. Future studies may be needed to incorporate the degrees of organ dysfunction, such as the sequential organ failure assessment (SOFA) score used in sepsis, into prognostic models for patients with FM on V-A ECMO.

In addition to analyzing data at specific time points, we explored the utility of dynamic change of organ functions in FM patients on V-A ECMO. In theory, end-organ damage is mitigated after ECMO support since V-A ECMO stabilizes circulation and improves tissue perfusion [22]. The progression of organ dysfunction despite ECMO support reflects inadequate tissue perfusion, despite the recovery of circulation, and may be suggestive of poorer outcomes. Mungan et al. [23] discovered that lactate clearance, before and after ECMO therapy, was superior to single lactate measurements as a prognostic sign of mortality. Slottosch et al. [22] also suggested that the dynamic course of lactate during ECMO therapy, compared to single lactate measurements, was predictive of 30-day mortality. In our study, we did not observe statistical differences in lactate clearance between the survivor and non-survivor groups, which may be highly possible due to the small sample size of our study.

Notably, within 24 h of ECMO support, non-survivors had creatinine increase by 108% and BUN increase by 46.4%, while survivors had creatinine decrease by 8.5% and BUN increase by only 3.9%. The ROC curve also showed an increase in serum creatinine by 68% within 24 h after ECMO support was associated with an increase in in-hospital mortality with an excellent AUC of 0.91. Acute kidney injury has been found to be one of the important prognostic factors in patients who received VA-ECMO for various indications [24]. Based on the above findings, measures to prevent kidney injury, such as avoiding hypotension and refraining from the use of renal-toxic agents, should be implemented at early phases of disease in patients with FM, even before ECMO support. The early change of renal function, before and after ECMO therapy, could be indicative for prognostication, and further extensive studies are required to confirm the finding. 

Percutaneous ventricular assist devices, such as Impella (Abiomed Inc., Danvers, MA, USA), are excellent tools that provide mechanical support while avoiding complications associated with ECMO. Several case reports [25,26,27] detail their successful experiences of using percutaneous ventricular assist devices in patients with FM complicating cardiogenic shock. A scientific statement from the American Heart Association acknowledges percutaneous ventricular assist devices as one of the feasible tools to support FM patients complicating cardiogenic shock [2]. Given the rare incidence of FM, it is challenging to design RCTs to compare clinical outcomes of FM using different modalities of mechanical circulatory support (e.g., ECMO versus Impella). In Taiwan, Impella devices are currently not available for commercial use. However, we believe percutaneous ventricular assist devices could be a reasonable alternative or adjunct to ECMO. Further studies are required to identify the populations that will benefit most from the devices.

To our knowledge, our study is the first to explore the implications of the dynamic change of organ functions in patients with FM on V-A ECMO. The study has several limitations. First, patients in our cohort did not receive a endomyocardial biopsy, which was the gold standard of diagnosing fulminant myocarditis. The biopsy results could also be useful to guide treatment in certain diagnoses, such as eosinophilic myocarditis. Nevertheless, we did apply strict criteria for establishing the diagnosis of FM during our chart review and all patients received cardiac catheterization to rule out ACS. Second, viral panels were not available since it was not routinely performed in our hospital. Third, studies may be underpowered to detect statistical differences due to the small sample size. Finally, the analysis of AST, ALT, or bilirubin were not available since there were six missing data in those variables.

## 5. Conclusions

V-A ECMO is an excellent tool to support patients with FM complicating cardiogenic shock. Worsening troponin I, CK-MB, and renal functions within 24 h of ECMO support were associated with an increase in in-hospital mortality. Measures to prevent kidney injury should be implemented at an early phase of FM, even before ECMO support. Future prospective studies are required to determine the role of the dynamic change of organ dysfunctions in prognostication in this population.

## Figures and Tables

**Figure 1 healthcare-10-01063-f001:**
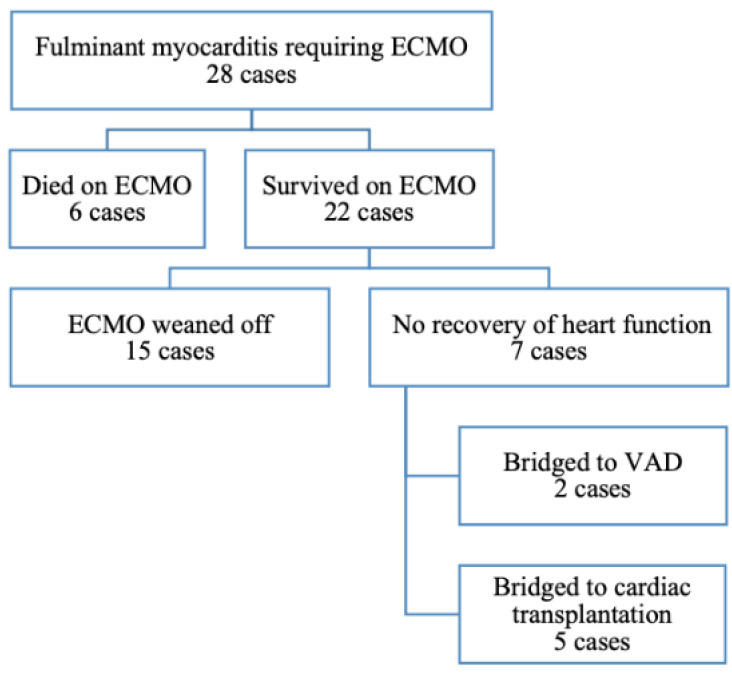
Study population. ECMO extracorporeal membrane oxygenation; VAD ventricular assist device.

**Figure 2 healthcare-10-01063-f002:**
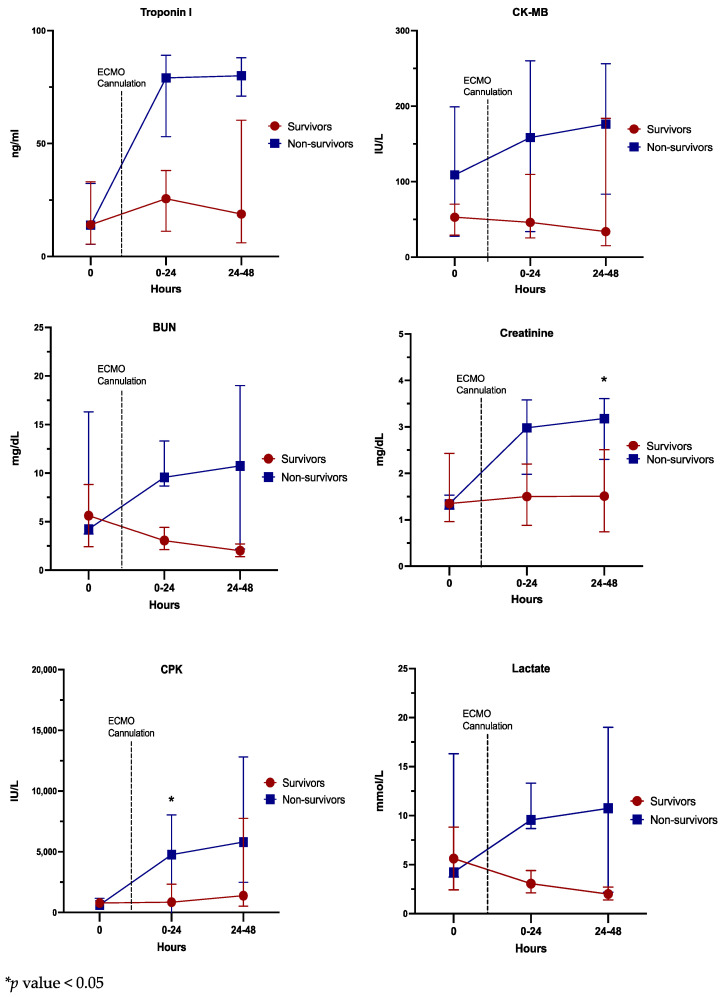
Laboratory data at different time points (24 h prior, 24 h after, and 24–48 h after ECMO support) among non-survivors and survivors.

**Figure 3 healthcare-10-01063-f003:**
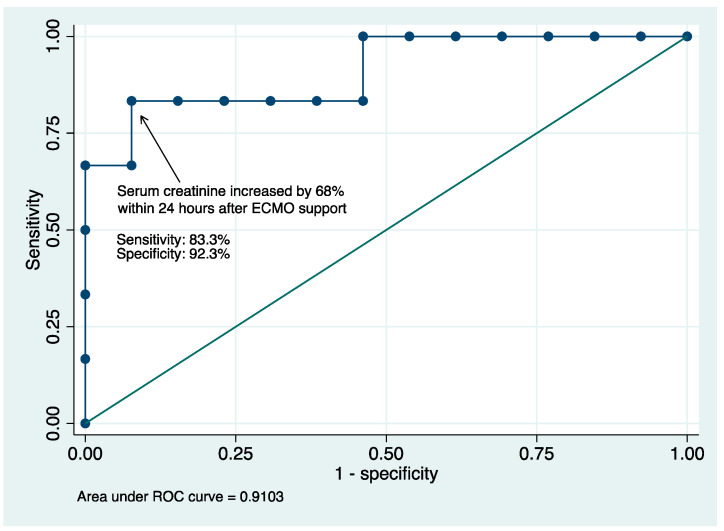
Receiver operating characteristic curve: Differences of serum creatinine within 24 h before and after ECMO cannulation in predicting in-hospital mortality.

**Table 1 healthcare-10-01063-t001:** Patient characteristics, SAVE score, LVEF, steroid/IVIG use, and laboratory data at baseline ^a^.

	Non-Survivors (*n* = 6)	Survivors (*n* = 22)	*p*-Value
Age, years	44.7 ± 18.6	49.2 ± 11.9	0.47
Female sex	3 (50%)	12 (54.5%)	1
Body mass index	23 (23–27.2)	23.9 (21.8–26.8)	0.9
Comorbidities			
Diabetes Mellitus	1 (16.7%)	2 (9.1%)	0.53
Non-occlusive coronary artery disease	1 (16.7%)	0	0.21
CKD stages 3–5	0	2 (9.1)	0.61
SAVE score	−4.3 ± 6.5	−1.4 ± 5.8	0.29
Time from ED triage to ECMO cannulation (min)	681 (140–1260)	315 (140–766)	0.73
Refractory ventricular tachycardia/fibrillation	0	2 (9.1%)	1
ECMO during CPR	1 (16.7%)	3 (13.6%)	1
LVEF (%) ^b^	19.9 (13.9–30.5)	23.1 (18–23.4)	0.92
Steroid use	1 (16.7%)	7 (31.8%)	0.39
Intravenous immunoglobin use	1 (16.7%)	1 (4.6%)	0.64
Laboratory data prior to ECMO support			
WBC (K/uL)	11.1 (8.2–14.2)	10.9 (8.6–14.8)	0.73
Hb (g/dL) ^c^	15.4 (13.1–15.9)	13.8 (12.6–14.8)	0.25
Platelet (K/uL) ^c^	201 (175–287)	185 (112–270)	0.58
CK-MB (IU/L)	108.9 (27.8–199)	52.9 (29.2–70.3)	0.56
Troponin I (ng/mL)	13.9 (5.5–32.3)	14.1 (5.5–33.1)	0.98
CPK (IU/L) ^c^	620 (274–1171)	788 (419–1096)	0.84
Lactate level (mmol/L)	4.2 (3.7–16.3)	5.6 (2.4–8.8)	0.67
BUN (mg/dL)	21 (21–28)	21 (14–33)	0.99
Creatinine (mg/dL)	1.34 (1.23–1.53)	1.35 (0.96–2.43)	0.88
Arterial PH	7.25 (7.14–7.38)	7.36 (7.3–42)	0.29
Peak CK-MB during hospitalization (IU/L)	185.9 (83.5–727.2)	70.3 (44.5–149)	0.16
Peak troponin I during hospitalization (ng/mL)	66.6 (16.6–82)	31.9 (10.9–47.6)	0.15

CKD: chronic kidney disease; CPR: cardiopulmonary resuscitation; ECMO: extracorporeal membrane oxygenation; ED: emergency department; LVEF: left ventricle ejection fraction; SAVE: survival after veno-arterial ECMO. ^a^ Bilirubin, lactate dehydrogenase, and central venous oxygen saturations were excluded for analysis since there were more than five missing data in total in survivors’ and non-survivors’ groups. Liver enzymes (AST and ALT) were not presented since there were 6 missing data in each variable. ^b^ One missing data in survivor group and one missing data in non-survivor group. ^c^ Two missing data in survivor group.

**Table 2 healthcare-10-01063-t002:** Clinical outcomes and incidence of major complications.

	Non-Survivors (*n* = 6)	Survivors (*n* = 22)	*p*-Value
Duration of ECMO (days)	4.5 (4–9)	8 (5–14)	0.1
ICU length of stay (days)	5 (4–9)	20 (11–45)	0.002 *
Renal replacement therapy	5 (83.3%)	10 (45.5%)	0.23
Major complications			
Limb ischemia	1 (16.7%)	0	0.21
Bowel ischemia	1 (16.7%)	0	0.21
Major GI bleeding	1 (16.7%)	4 (18.2%)	1
Major bleeding from cannula sites	0	1 (4.6%)	1
ECMO cannula malposition or dislodgement	0	0	N/A
Ischemic stroke or intracranial bleeding	0	0	N/A

ECMO: extracorporeal membrane oxygenation; GI: gastrointestinal; N/A: not applicable; ICU: intensive care unit. * *p*-value < 0.05.

**Table 3 healthcare-10-01063-t003:** Differences of variables within 24 h prior and after ECMO cannulation (Δ).

	Non-Survivors (*n* = 6)		Survivors (*n* = 22)		*p*-Value ^1^
	Δ	Δ in %	Δ	Δ in %	
WBC (K/uL)	1.1 (0 to 4.1)	6.9% (0 to 32.1)	0.6 (−0.7 to 2)	6.5% (−10 to 12.5)	0.7
Hb (g/dL)	−1.9 (−3.4 to −0.7)	−13.7% (−21.5 to −5.3)	−1.7 (−2.4 to −1.1)	−12.4% (−16.4 to −7.4)	0.82
Platelet (K/uL)	−92 (−138 to −47)	−38.4% (−52.5 to −16.4)	−49.5 (−76 to −8)	−28.3% (−37.6 to −1.9)	0.49
CK-MB (IU/L)	49.5 (30.6 to 251)	96.8% (23.3 to 144.1)	−10.2 (−20.4 to −1.6)	−23.7% (−41.5 to −10.4)	0.022 *
Troponin I (ng/mL)	42 (7.5 to 46.8)	378% (145 to 1359.9)	−0.1 (−1.6 to 15.8)	1.7% (−18 to 147.5)	0.032 *
CPK (IU/L)	1418 (486 to 18601)	214% (41.5 to 2265)	257 (−35 to 1160)	41.6% (−11 to 190.7)	0.1
Lactate (mmol/L)	−1.8 (−3.1 to 2.4)	−18.7% (−25.4 to 54)	−2 (−4.4 to 0.5)	−32% (−64.5 to 29)	0.44
BUN (mg/dL)	13 (10 to 13)	46.4% (33.3 to 61.9)	1.5 (−4.5 to 4)	3.9% (−23.6 to 18.5)	0.006 *
Creatinine (mg/dL)	1.7 (0.7 to 2)	108% (52.3 to 156.6)	−0.1 (−0.4 to 0.3)	−8.5% (−25.6 to 17.5)	0.005 *
Arterial PH	0.09 (−0.07 to 0.12)	1.22% (−0.9 to 1.7)	0.05 (−0.02 to 0.13)	0.67% (−0.3 to 1.8)	0.89

Δ: difference of variables between 24 h prior and 24 h after ECMO cannulation (Variables [Pre-ECMO]—Variables [Post-ECMO]); Δ in %: (Variables [Pre-ECMO]—Variables [Post-ECMO])/Variables [Pre-ECMO] * 100%. ^1^ Mann–Whitney U test was conducted to assess the difference of the change in variables between survivors and non-survivors. *p*-value was reported accordingly. * *p*-value < 0.05.

## Data Availability

The data presented in this study are available on request from the corresponding author. The data are not publicly available due to the hospital policy of the China Medical University Hospital. Patient data, even after proper deidentification, are prohibited to be published in the public dataset.

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
