# Peer review of "The Early Dynamic Change in Cardiac Enzymes and Renal Function Is Associated with Mortality in Patients with Fulminant Myocarditis on Extracorporeal Membrane Oxygenation: Analysis of a Single Center’s Experience"

_healthcare, 2022, doi:10.3390/healthcare10061063_

Round 1
Reviewer 1 Report
The authors investigated in this article entitled “The early dynamic change in cardiac enzymes and renal function is associated with mortality in patients with fulminant myocarditis on extracorporeal membrane oxygenation: Analysis of 4 single center’s experience” the trends in several biomarkers before and after ECMO support in patients with fulminant myocarditis. They found that worsening troponin-I, CK-MB, and renal function within 24 hours following the initiation of ECMO support were associated with incremental in-hospital mortality in patients with fulminant myocarditis. Several concerns have been raised.
- The clinical implication to predict mortality following the initiation of ECMO is unclear, because we cannot change any therapeutic strategy at this timing. Higher levels of troponin-I, CK-MB, and creatinine during ECMO supports in the non-survivors might not be so surprising.
- Did the authors consider to use Impella concomitantly?
- How did the authors decide to decide non-recovery and perform more intensified therapies including durable LVAD and heart transplantation? Some patients may be deceased following the explantation of ECMO. Explantation of ECMO does not always result in successful outcome. All survivors enjoyed favorable clinical outcomes thanks to ECMO alone or bridge therapies? In other words, the timing of bridge therapies might be another strong prognostic factor.
Author Response
Point 1: The clinical implication to predict mortality following the initiation of ECMO is unclear, because we cannot change any therapeutic strategy at this timing. Higher levels of troponin-I, CK-MB, and creatinine during ECMO supports in the non-survivors might not be so surprising.
Response 1: Thank you for the comment. We agree that this present paper would not change the management or the therapeutic strategy at the timing of ECMO initiation. Like many previous publications, this paper addresses the factors associated with prognosis in patients who have fulminant myocarditis receiving V-ECMO which may help physicians to predict outcomes and initiate goal of care discussions with patients’ families. As for troponin I and CK-MB, we found, surprisingly, that there was no difference in troponin I and CK-MB levels between survivors and non-survivors at baseline. However, survivors and non-survivors had different trajectories in the change of troponin I and CK-MB following ECMO support. Through this paper, we highlight the value of assessing not only the absolute static lab values, but also the dynamic changes of organ functions for clinical prognosis.
Point 2: Did the authors consider to use Impella concomitantly?
Response 2: Thank you for the comment. Impella (percutaneous left ventricular assist device) is an excellent tool providing mechanical circulatory support while avoiding the potential complications in association with ECMO. Several publications have reported the use of percutaneous left ventricular assist devices in patients with fulminant myocarditis. A Scientific Statement from the American Heart Association acknowledges the percutaneous bi- ventricular assist device being one of the feasible tools to provide mechanical circulatory support. Given the rare incidence of fulminant myocarditis, it is challenging to design a randomized controlled trial to compare the clinical outcomes of using different modalities of mechanical circulatory support (e.g. ECMO versus Impella) in this setting.
In Taiwan, we do not have the Impella system available in hospitals for mechanical circulatory support. We agree that it is important to discuss the utility of impella in patients with fulminant myocarditis.
We have revised our manuscript accordingly in the discussion section.
Point 3: How did the authors decide to decide non-recovery and perform more intensified therapies including durable LVAD and heart transplantation? Some patients may be deceased following the explantation of ECMO. Explantation of ECMO does not always result in successful outcome. All survivors enjoyed favorable clinical outcomes thanks to ECMO alone or bridge therapies? In other words, the timing of bridge therapies might be another strong prognostic factor.
Response 3: Thank you for the comment. We typically assessed a patient's heart recovery daily based on echocardiogram as well as arterial line waveform and had a 10 to 14 day waiting period for cardiac recovery. If a patient had no signs of cardiac recovery beyond this time period, we would initiate the pathway to pursue bridging therapy. While waiting for a donor heart or surgery for LVAD placement, patients could still have explantation of ECMO if signs of cardiac recovery were noted. Donor hearts are scarce, especially in Taiwan. Therefore, most patient will go through evaluation for LVAD placement for bridging therapy.
We agree that the explantation of ECMO does not always result in a successful outcome. In our cohort, there were no patients who required emergent LVAD or restarting of ECMO following explantation of ECMO. This phenomenon may be explained by the small sample size or that we may be overly cautious in the explantation of ECMO.
We agree the timing of bridging therapy might be a strong prognostic factor. Future large studies are needed to demonstrate the correlation between clinical outcomes and the timing of bridging therapy.
We have revised the manuscript accordingly.
Reviewer 2 Report
In this work, as cleraly suggested by the title, Ching-Lin Ho et al found that early dynamic change in cardiac enzymes and renal function is associated with mortality in patients with fulminant myocarditis on V-A ECMO.
These are my remarks:
Pag 2, line 50: I suggest to take out the phrase “..to review the current literature in this 50 matter.” because you discuss some studies in the Discussion section but it is not a review in the full sense of the term. On the other hand, having done this review work, I suggest you do a detailed review in another paper.
In the abstract and on Pag 1, line 37: you mean fAtal and not fetal, I suppose.
In conclusion I think it is a good work, clearly described and well done. For these reasons, I agree in its publication.
Author Response
Point 1: Pag 2, line 50: I suggest to take out the phrase “..to review the current literature in this 50 matter.” because you discuss some studies in the Discussion section but it is not a review in the full sense of the term.
Response 1: Thank you for the suggestion. We have removed this phrase from the discussion section in this paper.
Point 2: In the abstract and on Pag 1, line 37: you mean fAtal and not fetal, I suppose.
Response 2: Thank you for catching the typo. The word “fetal” has been changed to “fatal” in the abstract.
Reviewer 3 Report
Authors investigated V-A ECMO was an excellent tool to support patients with FM complicating cardiogenic shock, and reported that worsening troponin-I, CK-MB, and renal functions within 24 hours of ECMO support were associated with an increase in in-hospital mortality. This result can become one component to manage optimal treatment and monitoring. The reviewer has concerns. Please consider and answer as below.
- Please add the title at the top of each figure (Troponin I, CK-MB, Creatinine, BUN, Lactate, and CPK) and error bar of Figure 2. In addition, it is better to start 0 in Creatinine, BUN, and Lactate.
- Please show the ROC curve in CK-MB and Troponin I within 24 hours of ECMO support.
Author Response
Point 1: Please add the title at the top of each figure (Troponin I, CK-MB, Creatinine, BUN, Lactate, and CPK) and error bar of Figure 2. In addition, it is better to start 0 in Creatinine, BUN, and Lactate.
Response 1: Thank you for the suggestion. We have edited the figure 2 so that titles of variables are present at the top. We also add the error bar with median and IQR. We adjust the start time to 0.
Point 2: Please show the ROC curve in CK-MB and Troponin I within 24 hours of ECMO support.
Response 2: We have added the comparison of creatinine, CK-MB and troponin I ROC curve in Appendix A.
In this article, we would like to highlight the ROC curve specifically for creatinine since it has the highest AUC compared to troponin-I and CK-MB. We would also like to highlight the information of optimal threshold in the ROC curve. The inclusion of all 3 ROC curves for creatinine, CK-MB, and troponin-I in a single figure may look graphically confusing to the reader, and we made the decision to only include the ROC curve of the creatinine in the text. However, please see the attached file for the ROC curves of troponin I, CK-MB, and creatinine respectively for reference.

Round 2
Reviewer 1 Report
There are no further comments.